# Research on Approximate Spatial Keyword Group Queries Based on Differential Privacy and Exclusion Preferences in Road Networks

**Liping Zhang, Jing Li and Song Li \***

School of Computer Science and Technology, Harbin University of Science and Technology, Harbin 150080, China; zhangliping0730@hrbust.edu.cn (L.Z.); 1704010306@stu.hrbust.edu.cn (J.L.)
* Correspondence: lisongbeifen@hrbust.edu.cn

**Abstract:** A new spatial keyword group query method is proposed in this paper to address the existing issue of user privacy leakage and exclusion of preferences in road networks. The proposed query method is based on the IGgram-tree index and minimum hash set. To deal with this problem effectively, this paper proposes a query method based on the IGgram-tree index and minimum hash set. The IGgram-tree index is proposed for the first time to deal with the approximate keyword query problem in the road network. This index significantly improves the efficiency of calculating the road network distance and querying approximate keywords. Considering that spatial keyword group queries are caused by NP-hard problems with high time complexity, this paper proposes a data structure that uses the minimum hash set, which can efficiently search for the result set. To address the problem that the traditional spatial keyword group query does not consider user privacy leakage and the limitations of existing privacy protection techniques, this method proposes a differential privacy-based allocation method to better protect the privacy of data. The theoretical study and experimental analysis show that the proposed method can better handle the approximate spatial keyword group query problem based on its use of differential privacy and exclusion preferences in road networks.

**Keywords:** exclusion preferences; spatial keyword group query in road network; privacy protection

## 1. Introduction

As people use GPS devices more and more frequently in their daily lives and share geolocation information on social media platforms, a large amount of geotagged data are being collected and recorded. These data consist of not only geolocation information, but also related keywords or tags, such as restaurants, attractions, stores, and so on. Spatial keyword querying refers to finding the matching point-of-interest objects (POI, point of interest) in spatial–textual databases. This technique plays an important role in many fields, such as geographic information systems, image retrieval, smart cities, and communication systems. As a result of the research of many scholars, spatial keyword querying has also developed many query models, such as the spatial keyword nearest neighbor query problem [1,2], the top-k spatial keyword query [3–5], the inverse nearest neighbor query [6,7], and the spatial keyword group query [8–11]. The spatial keyword nearest neighbor query finds the nearest few POI objects in terms of spatial distance. The top-k spatial keyword query scores the most POIs according to a scoring function, resulting in the top-k POIs with the highest scores. The spatial keyword inverse nearest neighbor query returns a POI object that is closest in distance to multiple points. Each POI returned by the above query models must match all the keywords required by the query. However, a spatial keyword group query is a set of POIs that jointly match the keywords of the query. With the growing demand for spatial keyword group queries in daily life, traditional query models may not be able to provide accurate and practical results for a large number of

keyword combination queries. To meet this complex demand, it is especially important to study and improve spatial keyword group queries. For example, a challenging scenario arises when a pair of friends schedule a weekend gathering and expect to eat Chinese food, go shopping, or go to the park in the course of the day, but where one person does not eat pork. Therefore, when recommending suitable places, it is necessary to consider both the positive and the negative preferences of the user, to ensure that the results cover all needs. In recent years, spatial keyword group queries, also called collective spatial keyword queries, have gradually become more widely used in daily life and have attracted more and more scholars' attention and research. The concept of the m-closest keywords (mCK) query was first proposed in [12]. It finds m points of interest, and the keyword information of this set of points of interest jointly covers the query keywords and minimizes the maximum pairwise distance of the objects in the group. However, the particular algorithm proposed in [12] had low fitness for large datasets, so to improve the efficiency of mCK queries, Choi et al. [13] proposed an algorithm with an approximation factor close to 1.15, based on the farthest-color Voronoi diagram. Since the study by Choi et al. only focused on the case in which each object contains only one query keyword, Guo et al. [14], taking into account the possibility of multiple keywords, relaxed the constraint of having only one keyword per object and proposed an approximation algorithm with a factor of $(1.15 + \varepsilon)$. The traditional spatial keyword group query described above only focuses on the study of keyword constraints and distance constraints, and the database environments are all in ideal Euclidean space. However, for people's convenience in daily life, the spatial keyword group query needs to be better adapted to more realistic geographic environments such as map search and group wisdom-aware traffic networks. And these geographic spaces, also called road network environments, can be abstracted as graph data structures for research. Therefore, the spatial keyword group query in road network environments is more meaningful. Gao et al. [15] studied more specifically the spatial keyword group query problem in road networks and proposed two approximation algorithms with provable approximation bounds and an exact algorithm to efficiently support the collective spatial keyword query in road networks.

Currently, existing spatial keyword group queries only have keyword exact queries and distance constraints, without considering the needs of simultaneous user input bias and exclusion preferences, and mostly focus on Euclidean space. Therefore, this paper proposes an approximate spatial keyword group query problem based on differential privacy and rejection preference in road networks. To deal with this problem effectively, this paper proposes a query method, based on the IGgram-tree index and minimum hash set, which is called IGHashDP. The main contributions of this paper are as follows:

(1) Aiming at the traditional spatial keyword group query study that does not consider user input bias and user rejection preference, this paper proposes a new query model, namely the approximate spatial keyword group query problem in road networks based on differential privacy and rejection preference. This query model is based on the traditional group query study, which is more in line with the actual road network environment. And it also takes into account the user's input bias and rejection preference, which is more in line with the varied needs of users in today's society.

(2) Currently, existing indexing techniques cannot deal with the query model problem proposed in this paper, so this paper proposes a new type of index, the IGgram-tree. This index not only has the advantages of G-tree [16] but also introduces an n-gram index [17] and inverted file technology to handle keyword queries. In terms of handling exclusion preferences, the index introduces Bloom filters, which can efficiently handle exclusion keywords information. To improve the query efficiency, a filtering algorithm is further proposed, based on the IGgram-tree index. It uses the IGgram-tree to perform the first step of the pruning operation to derive the objects in the spatial database that meet the query requirements of keyword constraints and exclusion preferences, thus reducing the computational overhead of subsequent queries.

(3) To address the problem that the traditional spatial keyword group query algorithm does not perform data privacy protection, this paper proposes a protection algorithm based on differential privacy. It is based on differential privacy and privacy protection of the exact result of the query through an indexing mechanism, solving the problem of privacy leakage that may result from traditional methods.

Compared with the PQ of previous methods, operation efficiency is improved by 17% using the proposed method. The rest of this paper is organized as follows. Section 3 provides the relevant important definitions. Section 4.1 proposes a new index, the IGgram-tree index, and a filtering algorithm based on this index, and Section 4.2 proposes a refinement algorithm based on the minimum hash set. Section 5 of this paper proposes a differential privacy-preserving algorithm for query results. Section 6 contains the corresponding experimental analysis. Section 7 provides a summary. The specific process is shown in Figure 1.

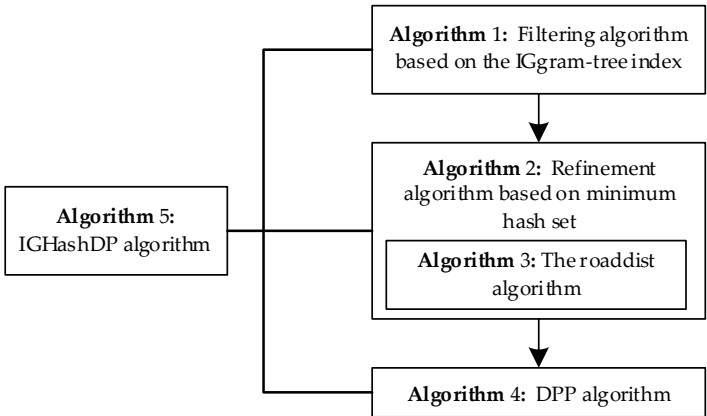

**Figure 1.** Algorithm relationship and data-processing flow.

## 2. Related Work

This section briefly summarizes the existing related research work and presents the research questions of this paper.

The spatial keyword group query has a wide range of applications in daily life, such as smart cities, smart recommendations, and other geographic information services. For example, a tourist who is visiting a city and planning the places he wants to visit from his hotel needs to include the activities he wants to do, such as eating, visiting the park, and watching movies for a period of time. Such decision-based query problems can obtain satisfactory results using spatial keyword group queries. Therefore, a large number of researchers have proposed a series of query models in recent years. Deng et al. [18] presented an optimal keyword coverage problem, a variant of the mCK query problem, which considers both intra-group distance and keyword weights and incorporates both factors into a linear cost function to propose an exact algorithm to solve the problem. Subsequently, Li et al. [19] studied spatial keyword group queries and proposed a parameterized approximation algorithm that allows the approximation ratio to be adaptive and the user to assign arbitrary query precision. However, the efficiency of the algorithm has not improved much in the process of increasing the accuracy.

Unlike querying in an ideal Euclidean space, some researchers have proposed spatial keyword group queries in a road network environment. Islam et al. [20] proposed a popularity-aware aggregated keyword in road networks, aiming to find a set of popular POIs (i.e., a popularity region). The POIs cover the query keywords and satisfy the distance requirement from each node to the query node and between each node pair, such that the sum of the scores of these nodes for the query keywords is maximized. To this end, scaling techniques for scoring were proposed to reduce the search space, and redundant computation reduction techniques were proposed to reduce the redundant computations in query processing. Su et al. [21] also studied ensemble spatial keyword queries in a road

network setting, but the difference is that this study is based on group-based ensemble queries, i.e., GBCK. Although all the above research works consider a set of keywords for querying, none of them consider the user's intention to reject.

Most of the current work investigating spatial keyword group queries is based on exact queries that do not take into account the possibility of input bias by users. However, in the field of spatial keyword querying, research work has been done on fuzzy queries. Hu et al. [22] investigated the top-k keyword fuzzy query problem on spatial data taking into account the fact that LBS (location-based services) systems cannot return relevant results when there are small differences between the query keywords and the underlying data. Zhang et al. [23] proposed a multi-spatial keyword fuzzy query algorithm. The algorithm converts the previous two-dimensional spatial distance calculation into Morton code matching to improve the query efficiency and uses the fuzzy matching algorithm to support query fault tolerance.

Efficient queries are important, but they involve network transmission during the query process or during the publication of results and thus face multiple privacy leakage risks. In order to protect data privacy, Yang et al. [24] used network Voronoi diagrams and some cryptographic primitives to address the privacy problem of simultaneously protecting spatial data and nearest neighbor queries. However, traditional privacy protection methods such as k-anonymity can be attacked with background knowledge, thus compromising privacy. Therefore, Dwork et al. [25] proposed using differential privacy techniques to better protect data. Since then, differential privacy protection methods have received much attention from researchers. For example, a classification transformation perturbation mechanism satisfying differential privacy was proposed [26]. The mechanism divides the transform range and segments the continuous numerical data, transforms the data according to the segments and perturbs them using a random response mechanism, and then randomly and uniformly selects the values from the segments identified by the perturbed data as the perturbed values. Chen et al. [27] combined the concept of differential privacy with the design of a bispectrum auction and used an exponential mechanism to select the clearing price for a bispectrum auction in which the probability was exponentially proportional to the correlation value and improved the mechanism in terms of the auction algorithm, the utility function, and the design of the buyer grouping algorithm.

With the development of social technology, users' needs are also increasing. For spatial keyword queries, users often hope not only to query the objects that satisfy the required keywords but also to exclude the objects that contain the user's rejection keywords in the query process. Therefore, the spatial keyword query with rejection keywords has considerable research value. This paper proposes an approximate spatial keyword group query problem based on differential privacy and exclusion preferences in road networks.

## 3. Problem Descriptions

Based on the content of the research and related technologies applied, this section provides the following basic definitions.

In this paper, a road network environment is abstracted as an undirected entitled graph denoted as $G = (V, E, W)$, where $V$ represents the set of vertices in the road network, each vertex $v \in V$ denotes a road intersection or the end of a road, $E$ represents the set of edges in the road network, $e_{ij} \in E$ denotes the road section between vertex $v_i$ and vertex $v_j$, $W$ represents the set of distances of edges, and $w_{ij} \in W$ denotes the distance of $e_{ij}$. A POI in a road network is a spatial–textual object denoted as $p = (p.e_{ij}, p.dist, p.K)$, where $p.e_{ij}$ is the edge where the POI is $e_{ij}$, where $i < j$ is assumed, where $p.dist$ is the distance of the POI from the vertex $v_i$ where it is located with a smaller edge ordinal number, and where $p.K$ is the set of keywords of the POI.

**Definition 1.** *AERGSKQ. Let P be a POI dataset, an approximate spatial keyword group query based on differential privacy and exclusion preferences in road networks (AERGSKQ) is denoted as q = (q.l, q.K+, q.K−), where q.l denotes the query location, q.K+ denotes the set of positive keywords*

*for the query, and q.K− denotes the set of exclusion keywords for the query, i.e., the exclusion preferences for the user query. The query q should return a set of POIs that are jointly closest to the query point location and internally compact, and the set of keywords of POIs that jointly cover q.K+ and have no intersection with q.K−.*

**Definition 2.** *Feasible set S. Given an AERGSKQ query q, let S be the feasible set of q. Then S satisfies the following conditions:*

*(1)   $q.K+ \subseteq \cup p_i.K_{p_i \in S}$, the keyword concatenation of objects in S can cover all the keywords in q.K+, and approximate matching is considered for keyword matching in this paper.*

*(2)   $\forall key_{key \in q.K-} \notin \cup p_i.K_{p_i \in S}$, no object in S can contain any of the exclusion keywords specified by query q.*

**Definition 3.** *The intra-group distance of the feasible set S.indist. Given a feasible set S, when there is only one object in S, the intra-group distance of the feasible set S.indist is 0; when the feasible set contains a set of POIs, the intra-group distance of the feasible set S.indist is the road network distance of the two most distant POIs in S. The distance described in this paper is the road network distance.*

**Definition 4.** *The distance between the feasible set and the query point Dist(q, S). Given an AERGSKQ with its feasible set S, the distance between the feasible set S and the query point q, Dist(q, S), is the distance between the query point and the object in the feasible solution set that is farthest away from the query point, i.e., $\mathrm{Dist}(q, S) = \max_{p \in S} dist(q, p)$.*

**Definition 5.** *The spatial distance cost of a feasible set S.cost. Given a feasible set S, where the spatial distance cost is denoted as S.cost, the formula is calculated as shown in Equation (1):*

$$S.\cos t = \alpha \times S.indist + (1-\alpha) \times \mathrm{Dist}(q, S) \tag{1}$$

The smaller the value of *S.cost*, the higher the possibility that its corresponding feasible set *S* becomes the final result set of AERGSKQ. Here, the distance cost is calculated in a linear way, which can be more intuitive to understand the size of the weights of both. $\alpha$ is a smoothing parameter to balance the compactness of the objects in the feasible set and the distance between the feasible set and the query point. Without prejudice to generality, and for the convenience of research, in this paper $\alpha = 0.5$ is taken, and the distance cost of a feasible solution set can be simplified as Equation (2):

$$S.\cos t = S.indist + \mathrm{Dist}(q, S) \tag{2}$$

To better explain the problem, the following serves as an example. A company organizes a day's outing for its employees, involving, swimming, lunch, singing, fishing, and other activities. However, some employees cannot go to a restaurant that serves pork, for personal reasons. At this time, the query requirements are in line with the query problem proposed in this paper. Specifically, the query location for the company's booking of the hotel, the forward keywords for the company's planned activities in that place, namely {swimming pool, KTV, restaurant, fishing}, the exclusion of the keyword for the {pork}. The query returns a set of POIs that must cover the set of forward keywords, and none of them can have the 'pork' label. And its intra-group distance is the furthest road network distance between two and two in the POIs of this set of results, and its distance from the query point is the furthest road network distance from the hotel in the POIs of this set of results.

## 4. Query Algorithm Based on the IGgram-Tree Index and Minimum Hash Set

The query algorithm based on the IGgram-tree index and minimum hash set proposed in this paper is effectively divided into two parts. The first part is the filtering algorithm based on the IGgram-tree index, which proposes a new IGgram-tree index for fast filtering of POI objects in the dataset. The second part proposes using a data structure with the minimum hash set for storing the filtered POIs and then refining the candidate set to get the result set according to this structure.

### 4.1. Filtering Algorithm Based on the IGgram-Tree Index

This section proposes a new index structure IGgram-tree based on G-tree for initial processing of the dataset and efficient screening of POIs that meet the query keyword requirements as candidate sets. Firstly, the definitions of the graph partition and graph bounders are proposed as shown in Definitions 6 and 7.

**Definition 6.** *Graph partition [16]. Given a graph G = (V, E, W), where V is the vertex set, E is the edge set of G, and W is the weight set of the edges in the graph G. A partition of G is a set of subgraphs, i.e., $G_i = (V_i, E_i, W_i)$. And Gi satisfies the following conditions:*

*(1)*    $\cup_{1 \leq i \leq n} V_i = V$;
*(2)*    $\forall i \neq j, V_i \cap V_j = \varnothing$; *and*
*(3)*    $\forall u, v \in V_i, if (u, v) \in E, then (u, v) \in E_i$.

**Definition 7.** *Borders [16]. Given a subgraph $G_i$ of G, a vertex $u \in V_i$ is called a border if $\exists (u, v) \in E$ and $v \notin V_i$. A subgraph $G_i$ is called a supergraph of another subgraph $G_j$ if $V_j \subseteq V_i$ and $E_j \subseteq E_i$.*

Since the road network environment is different from the traditional Euclidean space, the road network distance needs to be used in calculating the distance between POIs, and an example of a road network environment designed in this paper is shown in Figure 2.

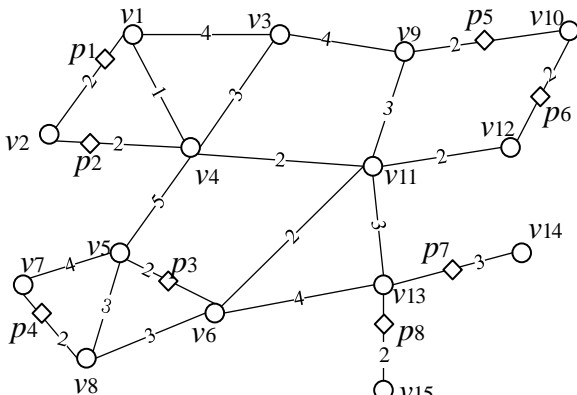

**Figure 2.** Illustration of road network environment.

As shown in Figure 2, each vertex in the road network represents a geographic location such as an intersection, each edge represents a road, such as a road or a bridge, and the number on the edge represents the weight of this edge, which represents the road distance. The POIs in the spatial–textual database *P* are distributed in a road network, denoted as $p_i$, on each edge $e_i$, and the spatial location information and keyword information of each POI object are shown in Table 1.

**Table 1.** Original information of the POIs.

| POI | Distance | Keywords |
|-----|----------|----------|
| $p_1$ | $(v_1,1)$ | $t1,t6$ |
| $p_2$ | $(v_2,1)$ | $t1,t2$ |
| $p_3$ | $(v_5,1)$ | $t1,t6$ |
| $p_4$ | $(v_7,1)$ | $t1,t3$ |
| $p_5$ | $(v_9,0)$ | $t6$ |
| $p_6$ | $(v_{10},2)$ | $t3$ |
| $p_7$ | $(v_{13},2)$ | $t5,t6$ |
| $p_8$ | $(v_{13},1)$ | $t5,t7$ |

The distance of the POI in Table 1 is expressed as the distance to the smaller endpoint of the edge on which it is located. For example, if $p_1$ is on edge $e_{12}$ in Figure 2, the distance is expressed as the distance to the endpoint $v_1$ as shown in Table 1 as $(v_1,1)$.

Since the calculation of road network distance is more complicated than the calculation of Euclidean spatial distance, efficient calculation of the road network distance determines the query efficiency to a certain extent. In this paper, we adopt the idea of the G-tree for graph partitioning, and the corresponding shortest distance is calculated and stored. Since the road network environment is fixed, the calculated shortest distance can continue to be used when the subsequent query conditions change, which greatly reduces the maintenance cost of the index. For this paper, considering that the user makes many errors in keyword Boolean matching due to input error, the n-gram index is incorporated for an approximation keyword query. In addition, an inverted file is created on each node to improve the efficiency of the keyword query. The graph partitioning of the road network according to Figure 2 is shown in Figure 3.

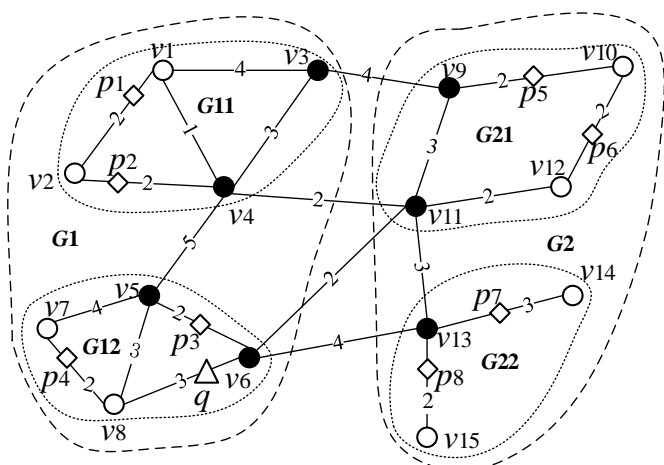

**Figure 3.** Illustration of road network environmental partition.

In Figures 2 and 3, the round points are the vertices of the road network, with the black round points being the borders of the partition, and the rectangular points are the POI points. The IGgram-tree index is established according to the obtained partitioned road network, as shown in Figure 4.

Each node of the IGgram-tree index represents a subgraph, the uppermost root node $G0$ represents the whole graph, and all subgraphs at the lower level are children of the root node. Each node contains the distance matrix, inverted n-gram file (IGF), Bloom filter file (BF), keyword intersection (UK), and boundary point number of that subgraph. The distance matrix of non-leaf nodes is the shortest distance matrix between the boundary points in this subgraph, and the distance matrix of leaf nodes is the shortest distance

matrix between the boundary points of this subgraph and all the vertices in the graph. By establishing the distance matrix, the data in the matrix can be read directly to calculate the distance of the road network in the smallest subgraph, so that only the distance between boundary points and the combination of them need to be considered when calculating the distance between two points in the whole road network environment, which significantly reduces the time cost.

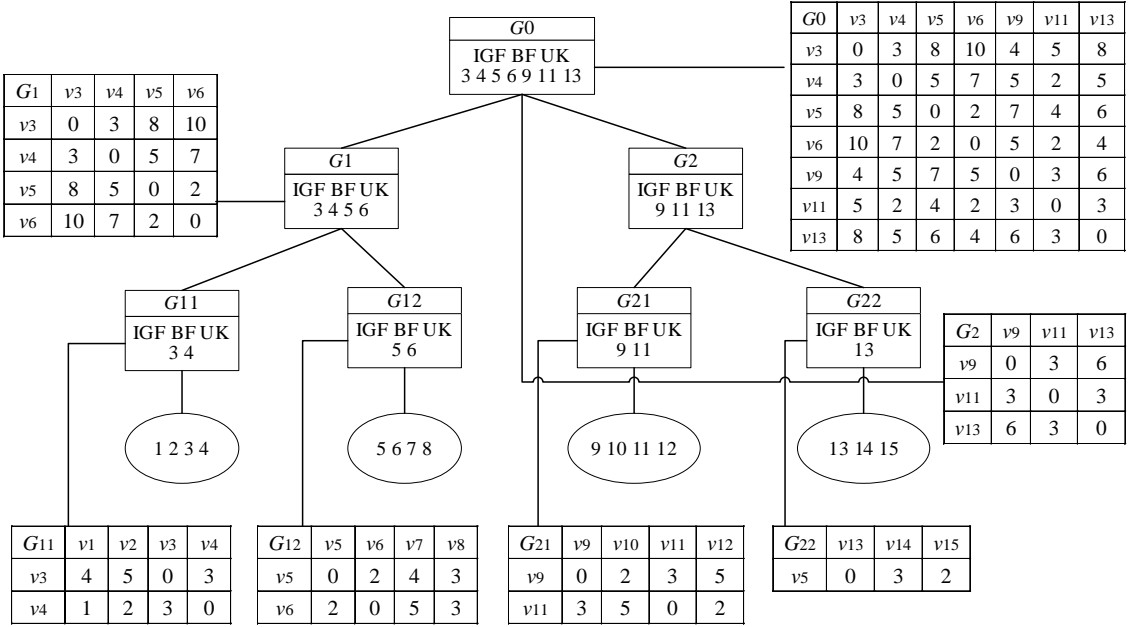

**Figure 4.** The IGgram-tree index.

In terms of keyword matching, the index introduces n-gram indexing based on inverted files. Specifically, the suffix of the n-gram content set of the keyword is added after the index keyword of each record in the inverted file, which can directly match the approximate query with the exact keyword in a one-to-one manner. Suppose the keyword information of POI objects $p_1$ and $p_2$ contains the keyword coffee, and the 3-g index for coffee is {$$c, $co, cof, off, ffe, fee, ee$, e$$}, then the row of records belonging to coffee in the inverted q-gram file has the format of coffee-{$$c, $co, cof, off, ffe, fee, ee$, e$$}: a sequence of graph partitions or POI objects containing coffee. Without loss of generality, the 3-g index is used in this paper in the IGF.

In terms of user exclusion preferences processing, the traditional spatial keyword query method with exclusion keywords establishes keyword dichotomous tree indexes on the spatial indexes according to the exclusion keyword content of the query, resulting in the need to rebuild the indexes every time the query requirements change, and the index maintenance cost of this processing method is too high. In contrast, the IGgram-tree index proposed in this paper establishes a Bloom filter for the keyword concatenation of POI objects in each partition, so that the existence of POI objects in the partition that do not meet the query requirements can be judged directly by its Bloom filter when determining the excluded keywords. Combined with the keyword intersection UK established for each node in the index, it is possible to determine whether the entire partition can be pruned directly. The proposed processing method can efficiently determine the excluded keywords and pruning, and quickly reduce the search space.

Based on the IGgram-tree indexing characteristics, this paper proposes the following two pruning theorems.

**Theorem 1.** *Given a spatial–textual database P and the corresponding IGgram-tree index and query q, note that a node in the index is G. If its Bloom filter G.BF determines that the keyword ekey appears in the set of excluded keywords q.K− results in existence, and the keyword intersection G.UK of node G contains a keyword ekey in the set of excluded keywords of the query, i.e., $\underset{ekey \in q.K-}{\exists}$ ekey ∈ G.UK, then the node ekey exists in the UK of all children of the node, and the node and its descendant nodes should be pruned.*

**Proof of Theorem 1.** Using the converse method, according to the working principle of the Bloom filter, assume that there is no exclusion keyword ekey in the sub-node of G and $ekey \notin \forall G_i.UK$, then ekey must exist after the hash function mapping bit 0, and because $G.UK = \cap G_i.UK$, then $ekey \notin G.UK$, contradicting the original conditions, and the proof is complete. □

**Theorem 2.** *Given a spatial–textual database P and the corresponding IGgram-tree index and query q, let the set of n-gram index contents corresponding to the keyword key be denoted as $g_{key}$, and note that a node in the index is G. The n-gram index part of the keyword corkey in the G.IGF of this node is $g_{corkey}$, and if $\left|g_{key}\right| - \left|g_{key} \cap g_{corkey}\right| > n$, then prune to exclude the keyword corkey record.*

**Proof of Theorem 2.** When the keyword key has an approximation error as an input error, missing or adding a letter, the inconsistent content of $|gkey|$ and $|gcorkey|$ is the number n. Therefore, when $\left|g_{key}\right| - \left|g_{key} \cap g_{corkey}\right| > n$, this indicates that the query keyword key entered by the user is too different from the POI keyword corkey, and the probability of not being the same word is very low. □

To summarize, Theorem 1 indicates that the current traversal node and its children nodes are judged as to whether they contain the exclusion keyword through the Bloom filter and the keyword intersection in the index in concert. And if both the Bloom filter and the intersection are judged to contain it, the node and its children nodes can be pruned in their entirety. Theorem 2 involves filtering from the degree of approximation of keywords, by judging the degree of difference between the keyword n-gram content of the current node and the query forward keyword n-gram content to determine whether the current node contains the keywords required by the query.

Based on Theorems 1 and 2, the filtering algorithm based on the IGgram-tree index is further proposed as shown in Algorithm 1.

Algorithm 1 first initializes the priority queue *PQ* to store the nodes to be recorded during the query (lines 1–2). If the *PQ* is not empty, the dequeued object at the head of the queue is assigned as *G*. If *G* is a non-leaf node (lines 3–5), first match the exclusion keyword in *q.K−* in *G.IGF* according to Theorem 2 and modify the corresponding content of *q.K−* to the exact keyword if it reaches the approximate threshold (lines 6–9). Further, according to Theorem 1, determine whether its child nodes meet the exclusion keyword requirements of the query using the Bloom filter and the keyword intersection together, and queue those that meet the requirements into the *PQ* (lines 10–20); if *G* is a leaf node, iterate through the POI objects in it, and deposit those that meet the keyword requirements into the hash table *MH* (lines 21–29). When the *PQ* is empty, output *MH* (line 30). The data in *MH* are indexed by the keyword *key* in *q.K+*, and the index value is the string formed by joining the POI objects containing *key* and their distance information. Suppose the object $p_3$ containing the keyword *key* is at a distance of 1 from the boundary point $v_5$ of the smallest partition where it is located and $v_6$ is at a distance of 1. Then, the information string processed is "$p_3$:1$v_5$–1$v_6$", *MH* uses the zipper method when storing conflicts, and the objects in the chain are sorted in positive order by the smallest distance.

---

**Algorithm 1:** Filtering algorithm based on the IGgram-tree index

---

**Input**: IGgram-tree index on *P*, query *q(q.l, q.K+, q.K−)*.
**Output**: Candidate hash table *MH*.
begin
1: Initialize the priority queue *PQ* to empty and the hash table *MH* to empty;
2: the root node G0 joins the team PQ;
3: while *PQ* is not empty then
4:  *G ← PQ*.dequeue(); /*Queue *PQ* queues out an element assigned to *G*/
5:  if *G* is a non-leaf node then
6:   for exclusion keyword *ekey* in *q.K−* then
7:    if *ekey* matches to *G.IGF* keyword *gkey* with approximate distance $\leq 3$ then /*theorem 2*/
8:     *ekey* in *q.K− ← gkey* in *q.K−*; /* Correction of keyword information*/
9:    end if
10:    if *G.BF* and *G.UK* determine the existence of *ekey* results then /*theroem1*/
11:     continue;
12:    else then
13:     for *key* in *q.K+* then
14:      if *key* matches to *G.IGF* keyword *gkey* with approximate distance $\leq 3$ then
15:       *key* in *q.K+ ← gkey*;
16:       *PQ*.enqueue(*G*);
17:      end if
18:     end for
19:    end if
20:   end for
21:  else then /*G is a leaf node*/
22:   for POI object *p* in *G* then
23:    if *p.K* contains keyword *key* in *q.K+* and no keyword *ekey* in *q.K−* then
24:     calculate the distance $p_{vi}$ of *p* from the distance matrix of the node and its boundary point $v_i$ in the partition, keeping its distance-related information;
25:     *MH*.add(*p*);
26:    end if
27:   end for
28:  end if
29: end while
30: return *MH*; /* Store processed p information in MH*/
end

---

### 4.2. Refinement Algorithm Based on Minimum Hash Set

To further refine the filtering result of Algorithm 1 to yield the final query result, this section proposes a refinement algorithm based on the minimum hash set, using the data within the minimum hash set MH, combined with the distance matrix in the IGgram-tree index, to calculate the optimal query result.

The minimum hash set MH is used as the data structure for storing the output results in Algorithm 1. It takes the hash table as the basic structure and stores the candidate POI objects in MH according to their corresponding query keywords, thus improving the operational efficiency of the final results of the query at this stage. The specific structure of the minimum hash set is shown in Figure 5.

As shown in Figure 5, assuming that the query keywords in *q.K+* are *t*1, *t*3, and *t*5, and the exclusion keywords in *q.K−* are *t*2 and *t*4, the POI objects comprising the output of Algorithm 1 are $p_1$, $p_3$, $p_4$, and $p_6$–$p_8$. After calculating the distance between each of these POI objects and its boundary point within the smallest partition, the distance and its corresponding boundary point will be processed into the form of a string as the suffix of the POI object. And the string is the index value at the corresponding keyword index of the smallest hash set MH. When storing conflicts, MH uses the zipper method to store

multiple objects in the chain table there, and to sort according to the calculated minimum distance. The specific situation of the POI output of Algorithm 1 is shown in Table 2.

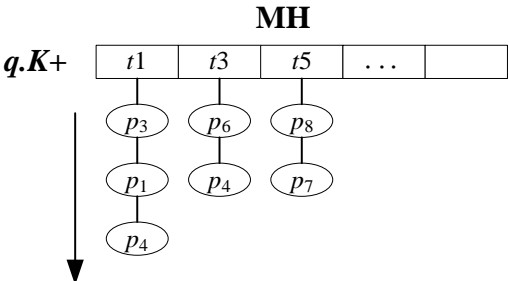

**Figure 5.** Illustration of MH's structure.

**Table 2.** Information after POI processing.

| POI | Bounders and Distances | POI | Bounders and Distances |
|------|------------------------|------|------------------------|
| $p_1$ | $v_4$:2, $v_3$:5 | $p_6$ | $v_9$:3, $v_{11}$:3 |
| $p_3$ | $v_5$:1, $v_6$:1 | $p_7$ | $v_{13}$:2 |
| $p_4$ | $v_5$:4, $v_6$:4 | $p_8$ | $v_{13}$:1 |

The distance information between POI objects containing a keyword in $q.K+$ can be obtained from the data in Table 2. Taking the keyword $t1$ as an example, the POI objects containing this keyword are $p_1$, $p_3$, and $p_4$. The processed information string of $p_1$ is "$p_1$:2$v_4$–5$v_3$", which means that the minimum distance from $p_1$ to the boundary point $v_4$ is 2, and the minimum distance from $p_1$ to the boundary point $v_3$ is 5. So, the minimum distance of $p_1$ at the boundary point of the smallest partition it is in is 2. Similarly, the minimum distance of $p_3$ at its minimum partition boundary point is 1, and the minimum distance of $p_4$ at its minimum partition boundary point is 4. Therefore, the order of $p_3$, $p_1$, and $p_4$ in the chain table at index $t1$ is arranged according to the minimum distance from smallest to largest, as shown in the direction of the arrow in Figure 5. Considering that this process often carries out the insertion of objects, the zipper part uses a chain table structure.

Since the graph partitions where the initial filtered POIs are located may be different, the distance from each POI to the query point still needs to be calculated. The proposed refinement algorithm based on the minimum hash set is shown in Algorithm 2, which can calculate the exact final result of the query.

Algorithm 2 performs the combinatorial analysis of feasible sets by sorting the optimized POI chains in the minimum hash set MH. First, initialize each intermediate result variable and result set; *S* is used to store a single feasible set, *Slist* is the set that stores all possible feasible sets, and the integer variable *temp* is used to store temporarily the road network distance between the query point *q* and its farthest POI object in *S*. For each possible feasible set stored in the *Slist*, the integer variable *min* stores temporarily the smaller *S.cost* value. Secondly, according to the contents of the POI chain table at different query keywords in MH, all feasible solution sets are generated and stored in *Slist* (lines 1–2), and each feasible set *S* is traversed to calculate its distance cost *S.cost*. The process always saves the content of the feasible set with the smallest cost and stores it in *Res* (lines 3–14) until *Slist* is empty and returns the last global result (line 15).

Algorithm 2 requires multiple calculations of the road network distance between two points. The process is complicated; the proposed roaddist algorithm is shown in Algorithm 3, which can efficiently calculate the minimum road network distance between two points in the road network according to the IGgram-tree index.

---

**Algorithm 2:** Refinement algorithm based on the minimum hash set

---

**Input**: Hash table *MH*, IGgram-tree index on *P*, query $q(q.l, q.K+, q.K-)$.
**Output**: Result set *Res*.
begin
1: Initialize the feasible set *S* to empty, the integers *temp*, min to 0, and the set *Slist* of feasible sets to empty;
2: take the Cartesian product of POI from its chain table according to different keywords in *MH* to form multiple feasible sets into *Slist*;
3: while *Slist* is not empty then
4:    *Slist* takes a feasible set and deposits it in *S*;
5:    for *p* in *S* then
6:      *temp* = max{temp, roaddist($p, rq$)};
7:      *S.indist* ← max{ roaddist($p_i, p_j$) | $i{\neq}j$};
8:    end for
9:    *S.cost* = *S.indist* + *temp*;
10:   if *min* > *S.cost* then
11:     *Res* ← *S*; /* Overwrite the new optimal set of feasible solutions into *Res**/
12:     *min* ← *S.cost*;
13:   end if
14: end while
15: return *Res*;
end

---

**Algorithm 3:** The roaddist algorithm

---

**Input**: IGgram-tree index on *P*, two POIs $p_1$ and $p_2$ in the road network.
**Output**: The shortest distance between $p_1$ and $p_2$.
begin
1: Locate the leaf node $G_{ij}$ where $p_1$ is located and the leaf node $G_{st}$ where $p_2$ is located;
2: initialize the integer variable *mindist* ← 0, $k$ ← 0, *curdist* ← 0, string *top* is empty, node $G_{cur}$ is $G_{ij}$;
3: if $G_{ij} = G_{st}$ then
4:    return DijkDist($p_1, p_2$);
5: else then
6:    $k$ ← Find the index of the first identical character of "*ij*" and "*st*";
7:    if $k < 0$ then
8:      *top* ← "0"; /* The common parent node of the two points is *G0* */
9:    else then
10:     *top* ← SubString($k$); /* The common parent of two points is the non-leaf node whose serial number is its common string, extracting the common string */
11:   end if
12:   while $G_{cur} \neq G_{top}$ then /* Traverse up from the leaf node until the common parent is queried and calculate the minimum distance path from $p_1$ to $G_{top}$ */
13:     *curdist* ← calculates the minimum distance from $p_1$ to the $G_{cur}$ bounder in the range of $G_{cur}$ based on its distance matrix;
14:     *mindist* ← *mindist* + *curdist*;
15:     $G_{cur}$ ← $G_{cur}$.parent; /* $G_{cur}$ is adjusted to its parent node */
16:   end while
17:   $G_{cur}$ ← $G_{st}$;
18:   while $G_{cur} \neq G_{top}$ then /* Traverse up from the leaf node until the common parent is queried and calculate the minimum distance path from $p_2$ to $G_{top}$ */
19:     *mindist* ← *mindist* + *curdist*;
20:     $G_{cur}$ ← $G_{cur}$.parent; /* $G_{cur}$ is adjusted to its parent node*/
21:   end while
22:   *curdist* ← Select the minimum distance according to the distance matrix of $G_{top}$'s child node bounders; /* Connect two paths to form the complete path from $p_1$ to $p_2$ */
23:     *mindist* ← *mindist* + *curdist*;
24: end if
25: return *mindist*;
end

The main idea of Algorithm 3 is to efficiently find the shortest distance path between two points and calculate the corresponding road network distance based on the graph partition and its distance matrix provided by the IGgram-tree index. First, locate the leaf node partition where the two POIs are located using the IGgram-tree index. If the two POIs are in the same minimum partition, i.e., in the same leaf node, the minimum road network distance is calculated directly using the Dijkstra algorithm (lines 1–4) because the subgraph data after the partition is small. Otherwise, the common parent node $G_{top}$ is calculated based on the analysis of the serial numbers of the leaf nodes where the two points are located (lines 5–11). Since the shortest distance paths between different partitions must pass through the bounders of the partitions on the way, the shortest distance paths are traversed upward from the leaf node partitions where $p_1$ and $p_2$ are located and find the shortest distance paths to the boundary points of their partitions, respectively, until both paths find the common parent $G_{top}$ (lines 12–21). Finally, the path from $p_1$ to $p_2$ is complete by connecting the last minimum distance path according to the distance matrix of $G_{top}$, adding up the path distance in the process, and returning the result to *mindist* (lines 22–25).

## 5. Differential Privacy-Based Protection Methods

After filtering and refining using Algorithms 1 and 2, the exact query results required by the user are filtered out. However, this is sometimes prone to privacy leakage, so this section proposes differential privacy techniques to encrypt the exact query results and protect the privacy of the data. This section first gives the definition of differential privacy and its important properties.

Differential privacy provides a way to balance privacy protection with data exploitation by adding noise or interference, randomizing the data to protect privacy, and ensuring that no personally identifiable or sensitive information is exposed during data distribution or analysis.

**Definition 8.** *ε-differential privacy [25]. Let there be a certain randomized algorithm A and let RA be the set consisting of all possible outputs of A. Given any two datasets P and P′ that differ by only one piece of data and any subset RRA of RA, A satisfies ε-differential privacy if algorithm A satisfies the following equation, i.e.:*

$$Pr[A(P) \in RR_A] \leq \exp(\varepsilon) \times Pr[A(P') \in RR_A] \tag{3}$$

The parameter $\varepsilon$ is called the privacy protection budget, which is used to control the degree of privacy protection, and the smaller $\varepsilon$ is, the higher is the degree of privacy protection.

**Definition 9.** *Global sensitivity [25]. With function f: $P \rightarrow A^d$, the global sensitivity of the function f is:*

$$\Delta f = \max_{p,p'} \|f(P) - f(P')\|_1 \tag{4}$$

For any dataset $P$ and $P'$ differing by only one piece of data, $\|f(P) - f(P')\|_1$ is the 1-order parametric distance between $f(P)$ and $f(P')$.

**Property 1.** *Sequence combination properties [28]. Let algorithms $A_1$, $A_2$, …, $A_m$ each satisfy $\varepsilon_i$-differential privacy ($1 \leq i \leq m$), and for dataset P, the sequence combination of algorithms {$A_1$, $A_2$, …, $A_m$} provides ε-differential privacy protection. This property indicates that the level of privacy protection is the sum of all privacy budgets when applying multiple differential privacy-preserving algorithms to the same dataset.*

**Property 2.** *Parallel combination property [28]. Let algorithms $A_1$, $A_2$, …, $A_m$ each satisfy $\varepsilon_i$-differential privacy ($1 \leq i \leq m$) for disjoint datasets $P_1$, $P_2$, …, $P_m$, and the parallel combination of algorithms {$A_1$, $A_2$, …, $A_m$} provides max $\varepsilon_i$-differential privacy. This property indicates that the level of privacy protection when applying the differential privacy-preserving algorithm to*

*multiple disjoint datasets is the lowest level of privacy protection among them, i.e., the maximum privacy budget.*

In differential privacy techniques, there are three protection mechanisms. For numerical data, the Laplace mechanism and the Gaussian mechanism are the most suitable. However, since the POI entity object of this paper is non-numerical data, the exponential mechanism is the most suitable. Therefore, the definition of the exponential mechanism is given below.

**Definition 10.** *Exponential mechanism [27]. Let the input of randomized algorithm A be a dataset P and the output be an entity object $r \in R$. The availability function is f (P, r), and $\Delta f$ is the sensitivity of the function f (P, r). If Algorithm A selects and outputs r from R with probability proportional to $\exp(\frac{\varepsilon f(P,r)}{2\Delta f})$, then Algorithm A provides ε-differential privacy.*

In order to achieve differential privacy, a suitable utility function f needs to be designed, as shown in Equation (5):

$$f(P,r) = |p_i|p_i, q \in G_{ij}| \tag{5}$$

The utility function $f$ indicates that the utility when the input dataset is $P$ and the output is $r$ is calculated as the number of POI objects in $r$ that are in the same partition as the query $q$. Therefore, it is known that the global sensitivity is 1.

According to the exponential mechanism, the probability distribution of the query results can be calculated according to Equation (6):

$$\Pr[A(f,P) = r] = \frac{\exp\left(\frac{\varepsilon f(P,r)}{2\Delta f}\right)}{\sum_{r' \in R} \exp\left(\frac{\varepsilon f(P,r')}{2\Delta f}\right)} \tag{6}$$

Based on the above analysis, Algorithm 4—the differential privacy preservation algorithm (DPP algorithm)—is proposed to protect the privacy of query results.

---

**Algorithm 4:** Differential privacy preservation algorithm (DPP algorithm)

---

**Input**: The precise result set Res($p_1, p_2, \ldots, p_n$) and the spatial–textual database *P*.
**Output**: Global results after protection *SafeRes*.
begin
1: Initialize *SafeRes* to empty, $r \leftarrow$ Res;
2: for $p_1$ to $p_n$ in $r$ then /* Iterate through each POI in the exact result set */
3:     $P_i \leftarrow$ random($p_i \in P$ && $p_i \neq p_1$–$p_n$); /* Generate a random POI that is not in Res */
4:     Replace the current object with $p_i$ to form a new set $r_i$;
5: end for
6: for $r$ and $r_1 \sim r_n$ then /* Use the exponential mechanism for Res and the generated proximity result set and save the output to the safe result set *SafeRes* */
7:     $\Pr[A(f,P) = r] \leftarrow \frac{\exp\left(\frac{\varepsilon f(P,r)}{2\Delta f}\right)}{\sum_{r_i \in R} \exp\left(\frac{\varepsilon f(P,r_i)}{2\Delta f}\right)}$ ;
8: end for
9: *SafeRes* $\leftarrow$ A(f, P, r);
10: return *SafeRes*;
end

---

Algorithm 4 performs differential privacy protection for the exact query results output by Algorithm 2. First, according to each POI object in the exact result set Res, an object $p_i$ different from $p_1$–$p_n$ is randomly selected from $P$ to replace the current object to generate the proximity dataset $r_1$-$r_n$ that differs by only one piece of data (lines 1–5), and then an index protection mechanism is added to the exact result set and the proximity dataset at a time to save the global results into *SafeRes* and output them (lines 6–10).

By this stage, the approximate spatial keyword group query problem based on differential privacy and exclusion preferences in the road network has been completed using the IGgram-tree index-based filtering algorithm, the minimum hash set-based refinement algorithm, and the differential privacy-preserving algorithm—that is, using the IGgram-tree index-based and minimum hash set-based query algorithm (IGHashDP) proposed in this paper. The complete algorithm is shown in Algorithm 5.

---

**Algorithm 5:** Query algorithm based on the IGgram-tree index and minimum hash set (IGHashDP algorithm)

---

**Input**: IGgram-tree index on *P*, query *q(q.l, q.K+, q.K−)*.
**Output**: Global results after protection *SafeRes*.
begin
1: Filtering algorithm based on the IGgram-tree index (Algorithm 1);
2: Refinement algorithm based on the minimum hash set (Algorithm 2);
3: Differential privacy preservation algorithm (DPP algorithm) (Algorithm 4);
end

---

The IGHashDP algorithm protects differential privacy for exact query results, but there is no conclusive answer to whether the IGHashDP algorithm achieves the required degree of differential privacy protection, so a privacy analysis of the IGHashDP algorithm is needed. The privacy of the algorithm is mainly reflected by the difference before and after processing the dataset. Based on the theoretical basis provided in the literature [27], Theorem 3 is proposed to prove that the algorithm satisfies $\varepsilon$-differential privacy.

**Theorem 3.** *IGHashDP algorithm satisfies $\varepsilon$-differential privacy.*

**Proof of Theorem 3.** Let the set of nearest neighbor outcomes be denoted by P and P′, Algorithm 3 be denoted by A, and RA denote the set of possible outcomes output by Algorithm A. Assume that the domain of values RA of the exponential mechanism is finite; however, this assumption is not necessary for the conclusion that for any r ∈ R the ratio of probabilities. □

$$\frac{\Pr[A(f,P)=r]}{\Pr[A(f,P')=r]}$$

$$= \frac{\frac{\exp\left(\frac{\varepsilon f(P,r)}{2\Delta f}\right)}{\Sigma_{r_i \in R_A} \exp\left(\frac{\varepsilon f(P,r_i)}{2\Delta f}\right)}}{\frac{\exp\left(\frac{\varepsilon f(P',r)}{2\Delta f}\right)}{\Sigma_{r_i \in R_A} \exp\left(\frac{\varepsilon f(P',r_i)}{2\Delta f}\right)}}$$

$$= \left(\frac{\exp\left(\frac{\varepsilon f(P,r)}{2\Delta f}\right)}{\exp\left(\frac{\varepsilon f(P',r)}{2\Delta f}\right)}\right)\left(\frac{\Sigma_{r_i \in R_A} \exp\left(\frac{\varepsilon f(P',r_i)}{2\Delta f}\right)}{\Sigma_{r_i \in R_A} \exp\left(\frac{\varepsilon f(P,r_i)}{2\Delta f}\right)}\right)$$

$$\leq \exp\left(\frac{\varepsilon}{2}\right)\left(\frac{\Sigma_{r_i \in R_A} \exp\left(\frac{\varepsilon}{2}\right)\exp\left(\frac{\varepsilon f(P,r_i)}{2\Delta f}\right)}{\Sigma_{r_i \in R_A} \exp\left(\frac{\varepsilon f(P,r_i)}{2\Delta f}\right)}\right)$$

$$\leq \exp\left(\frac{\varepsilon}{2}\right)\exp\left(\frac{\varepsilon}{2}\right)\left(\frac{\Sigma_{r_i \in R_A} \exp\left(\frac{\varepsilon f(P,r_i)}{2\Delta f}\right)}{\Sigma_{r_i \in R_A} \exp\left(\frac{\varepsilon f(P,r_i)}{2\Delta f}\right)}\right)$$

$$= \exp(\varepsilon)$$

Similarly, we can obtain $\exp(-\varepsilon) \leq \frac{\Pr[A(f,P)=r]}{\Pr[A(f,P')=r]}$, so the IGHashDP algorithm satisfies $\varepsilon$-differential privacy and can avoid privacy leakage of the results.

## 6. Experiment Analysis

This paper proposes an approximate spatial keyword group query method based on differential privacy and exclusion preferences in the road network, in view of the fact that the existing spatial keyword group query problem does not consider the approximate keyword query problem with exclusion preferences in the actual road network environment and does not address the user privacy leakage problem. The method proposed in this paper is divided into, firstly, a filtering and refining process for data points, and, secondly, adding differential privacy protection to the exact query results. To evaluate the performance of the method, five aspects of experiments are designed in this section. The first aspect compares the effect of the number of query positive keywords on the efficiency of different algorithms; the second aspect compares the effect of the number of query rejection keywords on the efficiency of different algorithms; the third aspect compares the accuracy of different algorithms executing results on different datasets; the fourth aspect compares the output probability of different results under different privacy budgets to judge the algorithm usability; and the fifth aspect compares the impact of different privacy-preserving algorithms on the accuracy of the result data at the time of publication. The methods that are compared with the proposed method in this paper are the SW algorithm [15] and the PQ algorithm [29].

The environment used for the experiments is Microsoft Windows 10 (64-bit), Core(TM) i7-7500U CPU@2.70 GHz processor, running memory of 12 GB, and programming language Java. Two real datasets are used for the experimental data, CAL (Cities of California) and TG (San Joaquim County). The CAL dataset contains 21,048 vertices and 21,693 edges, and the TG dataset contains 18,257 vertices and 18,263 edges. In this paper, we randomly generate 85,764 POI points for the CAL dataset and 24,264 POI points for the TG dataset, and we generate 1–5 keywords for each POI object from the keyword set of the dataset to which it belongs, with an average number of 2.5 keywords.

**Experiment 1.** *This part of the experiment aims to compare the impact of different algorithms on the algorithm efficiency in terms of the number of query positive keywords. Specifically, for each dataset, a certain number of query positive keywords are randomly generated from the keyword information of the dataset to which they belong as query positive keywords whose number variation interval is [1–5]. The CPU execution time variation and the number of extended nodes for the three algorithms are shown in Figure 6. Figure 6a,b show the influence on algorithm execution time and number of extended nodes for the CAL dataset of varying numbers of query positive keywords, and Figure 6c,d show the execution results of the algorithm in the TG dataset.*

Figure 6 shows that the execution efficiency of the proposed IGHashDP algorithm is always higher than that of the SW and PQ algorithms on both datasets as the number of query positive keywords increases. Since the IGHashDP algorithm uses a hybrid indexing technique to process and save the spatial and textual information in the database at one time, and although this consumes some time upfront, subsequent queries can be indexed directly. Therefore, the running time increases more slowly than the comparison algorithms. The PQ algorithm uses grid space indexing to simply partition the road network environment but does not store the distance-related information of the road network, which requires a complex graph search algorithm each time when calculating the distance, which means that the time consumption is larger. The SW algorithm stores the neighboring information of the road network, but does not combine the text information, so the calculation efficiency is lower, whether calculating the distance or filtering the text. The IGHashDP algorithm finely partitions the road network, the PQ algorithm uses a grid index that skews the data, and the SW algorithm has only basic adjacency information; therefore, as the query positive keywords increase, the number of extended nodes is the highest for SW, the second highest for PQ, and the lowest for IGHashDP.

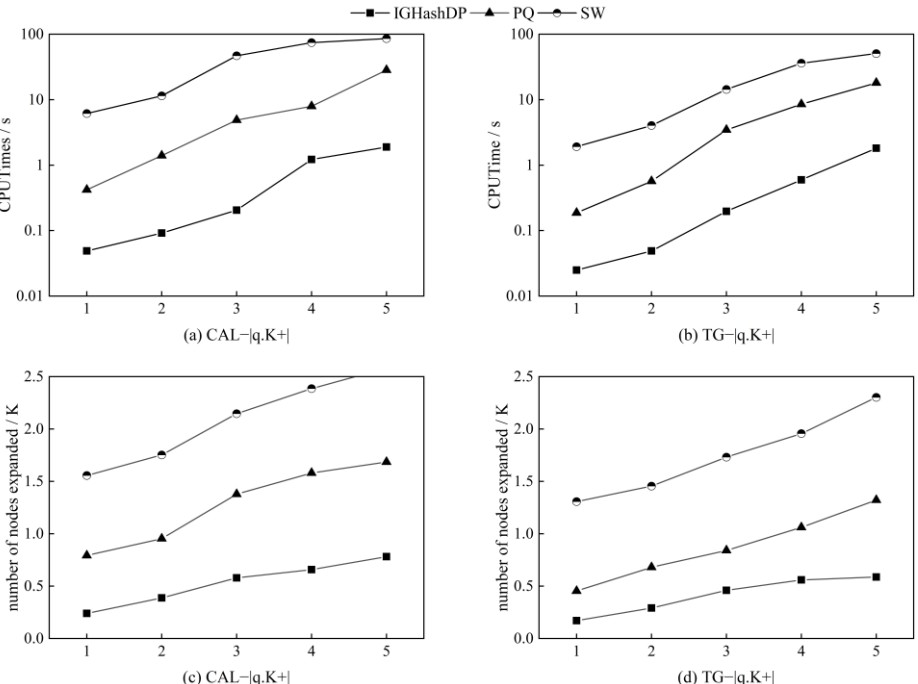

**Figure 6.** Effect of the number of query positive keywords on the efficiency of the algorithm and the number of expansion nodes.

**Experiment 2.** *This part of the experiment aims to compare the effect of different numbers of query rejection keywords on the efficiency of various algorithms. Specifically, for each dataset, a certain number of keywords are randomly generated from the keyword information of this dataset as rejection keywords, whose number varies in the interval [1–5]. In the cases in which the other algorithms do not consider rejection keywords, a keyword dichotomy tree is added for them. The CPU execution time variation and the number of extended nodes for the three algorithms are shown in Figure 7. Figure 7a,b show the algorithm execution time and the number of extended nodes for the CAL dataset as affected by the number of query rejection keywords, and Figure 7c,d show the execution results of the algorithm on the TG dataset.*

　　Figure 7 shows that the IGHashDP algorithm exhibits better operational efficiency than the SW and PQ algorithms as the number of query rejection keywords increases. Since the IGHashDP algorithm mainly uses Bloom filters to process the rejection keywords, it is efficient. And because of early pruning to reduce the search space, when the number of rejection keywords increases, the result set is smaller, its running time decreases, and the number of traversed nodes is smaller. In contrast, the traditional keyword dichotomous tree processing method needs to reconstruct all the indexes when the query content changes, so it is inefficient and needs to traverse to the subtree area of a keyword specifically when filtering the rejection keywords, which means that the number of extended nodes is larger.

**Experiment 3.** *The purpose of this part of the experiment is to compare the query accuracy of different algorithms on each dataset, and specifically for each dataset when the three algorithms are applied with the same number of query keywords and the same other metrics. The accuracy of algorithm execution is shown in Figure 8.*

　　Figure 8 shows that the accuracy of the IGHashDP algorithm outperforms the SW algorithm and the PQ algorithm on both datasets, remaining at least above 80%. Since the q-gram technique and the inverted file technique used in this paper perform exact lookups based on approximate queries, the accuracy cannot reach 100% due to the setting of the q-gram threshold and the existence of false positives in the Bloom filter. The SW algorithm performs the exact search based on the approximate algorithm, and therefore has the lowest accuracy.

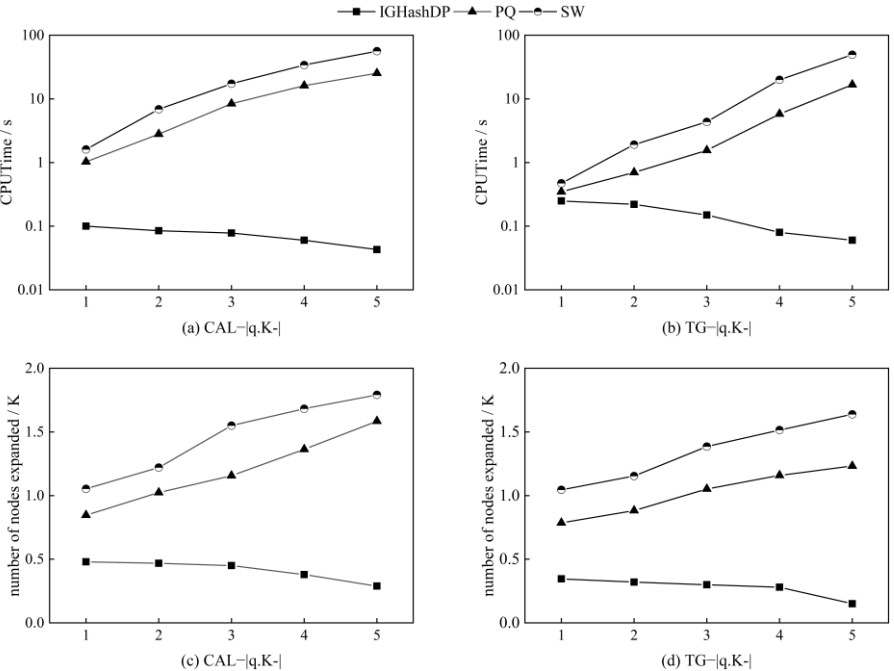

**Figure 7.** Effect of the number of query rejection keywords on the efficiency of the algorithm and the number of expansion nodes.

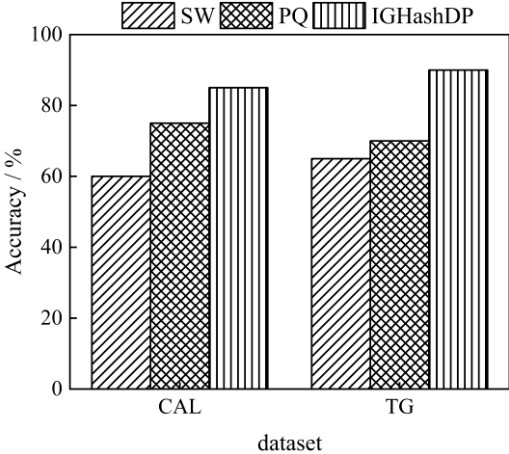

**Figure 8.** Algorithm accuracy.

**Experiment 4.** *This part of the experiment aims to compare the output probability of the correct outcome set with that of the confusion outcome set under different privacy budgets. The TG dataset is used for the experiment so that the privacy budget $\varepsilon \in \{0, 0.2, 0.4, 0.6, 0.8, 1\}$, and the average output probability $Mean(r_i)$ of the correct outcome set Res compared with the confusion outcome set $r_i$ is shown in Figure 9.*

Figure 9 shows that the probability of the correct result Res being output after the exponential mechanism for privacy protection is applied increases to a maximum of about 0.97, while the probability of the obfuscated result being output after the exponential mechanism is applied decreases to $10^{-5}$ orders of magnitude. Therefore, when $\varepsilon$ is large (e.g., $\varepsilon = 1$), the probability of the best result Res being output is increased, and when $\varepsilon$ is small, the difference in usability between the results is equalized; the degree of equalization increases as $\varepsilon$ decreases, and the corresponding output probabilities tend to be equalized as it decreases.

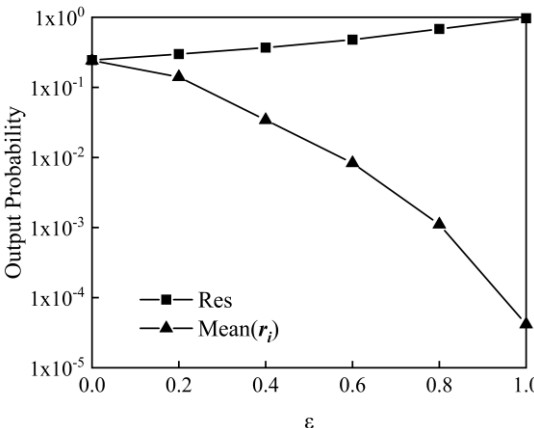

**Figure 9.** Availability comparison.

**Experiment 5.** *This part of the experiment aims to compare the impact of the change in the number of query keywords under different privacy-preserving methods on the accuracy of the experimental results when the data are published. The number is the sum of positive and rejection keywords, the range is taken as [2–6], the dataset used is CAL, and the experiment compares the IGHashDP algorithm with the Cloaking Region algorithm and the improved SecureKnnQuery protocol to protect the privacy query algorithm. The accuracy of the algorithms on the published results is shown in Figure 10.*

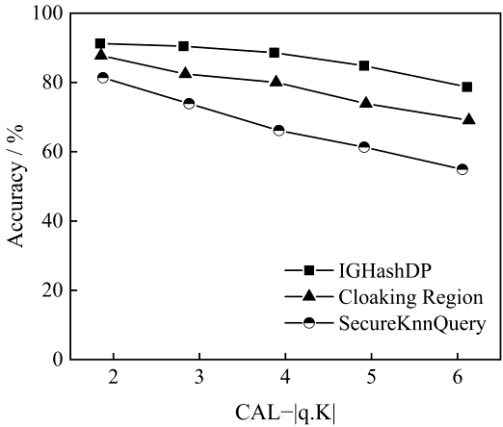

**Figure 10.** Impact of the number of keywords under different privacy-preserving methods on accuracy.

Figure 10 shows that the accuracy of all three algorithms decreases as the number of query keywords increases, but the accuracy of the IGHashDP algorithm is higher than that of the SW and PQ algorithms. Since the IGHashDP algorithm uses differential privacy techniques to protect the privacy of the results, the technique itself has higher accuracy than other protection techniques. The SecureKnnQuery algorithm [30] uses cryptographic primitives and involves server-side and user-side computations, but the complexity of the server-side computation encryption is high, and the number of iterations increases significantly as the number of query keywords increases, resulting in lower query accuracy. The Cloaking Region algorithm [24] uses the anonymous box technique, which leads to the situation that the nearer POIs are not traversed, so the query accuracy is low. In summary, the use of differential privacy techniques is more guaranteed to produce accurate results.

## 7. Conclusions

To address the existing spatial keyword group query problem that does not consider the problem of approximate keyword query with exclusion preferences in the actual road network environment and does not consider the user privacy leakage problem, a new

approximate spatial keyword group query based on differential privacy and exclusion preferences in road networks is proposed. This query is used to obtain a set of POIs that accurately match the user's desired keywords and exclusion preferences, and which are optimal in terms of distance when user input bias is taken into account in a road network environment. To deal with this problem effectively, this paper proposes a query method based on the IGgram-tree index and minimum hash set. First, a new index structure is proposed based on G-tree, which introduces Bloom filter technology and q-gram index technology for fast textual information judgment, and which also incorporates the advantages of G-tree for fast spatial distance information calculation. Thus, the filtering algorithm and corresponding refinement algorithm based on the IGgram-tree index are proposed. Then, to protect the data from privacy leakage, the query results are protected by the index mechanism of differential privacy. The experimental results show that the algorithm proposed in this paper has good scalability and efficiency, and the paper concludes with a recommendation for future research work focusing on the following aspects:

1. Streaming data-based spatial keyword group queries in dynamic environments.
2. Research on spatial keyword group query in the big data environment.
3. Approximate spatial keyword group queries based on user preferences in road networks.

**Author Contributions:** Conceptualization, Liping Zhang and Jing Li; methodology, Liping Zhang and Jing Li; investigation, Liping Zhang; writing—original draft preparation, Jing Li; writing—review and editing, Jing Li, Liping Zhang and Song Li; project administration, Song Li. All authors have read and agreed to the published version of the manuscript.

**Funding:** This research was funded by the National Natural Science Foundation of China, grant number 62072136, Key R&D Plan Project of Heilongjiang Province: 2022ZX01A34, and the National Key R&D Program of China, grant number 2020YFB1710200.

**Data Availability Statement:** The data presented in this study are available on request from the corresponding author.

**Conflicts of Interest:** The authors declare no conflict of interest.

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
