# Peer review of "Research on Approximate Spatial Keyword Group Queries Based on Differential Privacy and Exclusion Preferences in Road Networks"

_ijgi, doi:10.3390/ijgi12120480_

Round 1
Reviewer 1 Report
Comments and Suggestions for Authors
This paper studies an approximate spatial keyword group problem and proposes a query method based on the IGgram-tree index and the minimum hash set. Experimental results look good. But I have some suggestions as follows.
1. Why do we need to consider the user input bias and user rejection preference in a spatial keyword group query problem? Please try elaborating on its great importance or significance using real-life examples in Section 1.
2. The title ‘Definitions and Symbol Descriptions’ of Section 3 is not accurate, since this section does not contain all the definitions and symbol descriptions in the full manuscript. ‘Problem Descriptions’ may be more suitable here. And based on this, the authors are suggested to reconstruct Section 3.
3. The notation system throughout the manuscript is very confusing. After a symbol has been defined, its upper and lower case, superscript and subscript, orthography or italicization are all determined and cannot be modified arbitrarily in the course of use. Moreover, the same symbol has different sizes in different places in the text. If possible, it is recommended to use the LaTeX system to write the whole text.
4. There are many language-related problems. For example, the comma in the first paragraph of Section 4 is wrongly used to connect multiple sentences without conjunctions. It is recommended to check and revise the text with the help of a native English speaker.
5. Figures 1 and 2 are confusing. The authors are suggested to make some changes, such as removing some unnecessary marks, symbols, numbers, etc. and using colors to distinguish different areas. At the same time, the information conveyed by the figures needs to be explained clearly in words.
6. The relationship between Algorithms 1-5 is not clear. It would be better to give an overview of the whole algorithm in the form of pseudo-code or diagrams.
7. It is recommended to increase the space at the beginning of each line in the pseudo-code to make the structure of the algorithm clearer.
8. There is an error in the input of Algorithm 2. "Algorithm 1 outputs" should be removed.
9. The placement of the legends in the four sub-figures in Figure 5 is not consistent. And, if the legends of the four sub-figures are the same, you can keep only one of them.
Comments on the Quality of English LanguageExtensive editing of English language is required.
Reviewer 2 Report
Comments and Suggestions for Authors
Dear colleagues,
thank you a lot for your contribution on "approximate spatial keyword group queries based on differential privacy and exclusion preferences". For my point of view your work is highly relevant, especially the processing of exclusion mechanisms for preferences- and privacy control.
In the abstract and the introduction, you try to introduce into the need of an approximate keyword query with exclusion preferences in an actual road network environment. For my understanding theses sections are hard to follow, especially for the geospatial domain, which is not completely involved in natural language- and keyword processing. The introduction should try to explain the problem from the viewpoint of geoinformation, the road network and its relation to the keywords.
In the section related work a lot of challenges and their appropriate algorithms are listed. The direct relevance to the question of the paper was sometimes unclear and could be improved.
Section 3 highlights 5 definitions with their symbolization. Some explanation is given on the conditions of keywords and their spatial relation. This section is a core information of the paper as it delivers the basic algorithmic understanding. The definitions and conditions should be extended with some real-world examples in order to catch the reader more intensively.
Section 4 highlights the query algorithm based on the IGgram-tree index and a minimum hash set. Again this section delivers a definition for borders, the characteristics of a road network environment and a set of theorems. Although all theorems come along with a proof of theorem, its correctness is not supported by any cross-check or detailed explanation. This may be enough for mathematically inclined readers, but maybe not for the broad readership of IJGI.
Section 5 focuses on differential privacy-based protection methods, which list three further definitions, two properties and an additional theorem concerning IGHashDP for epsilon-differential privacy.
Section 6 elaborates on four experiments. The source of data, its grams or the relation to spatial characteristics is unclear for me. Maybe I have not understood the method of approximate spatial keyword group query, or the algorithms. It would be helpful to have a short introduction on the underlying methodology, respectively a simple definition of privacy budgets and their influence on preserving or excluding algorithms.
The conclusions introduce some new notions without reference. The references to G-tree, q-gram, IGgram should be added.
At least, what is the main proble, which has to be solved effectively. A short, very simple description of the problem could be helpful in the beginning and as resume in the conclusions.
Please excuse my possible lack of understanding n the name of readers of IJGI.
Best regards
Comments on the Quality of English Language
Text constellations and ways of expession should be checked professionally.
Reviewer 3 Report
Comments and Suggestions for Authors
Thank you for submitting your work. My overall critique is as follows:
Abstract
Line 9-12: The statement, “To address the existing spatial keyword group query problem ………………and exclusion preferences in road networks is proposed” is too long and difficult to understand. It is suggested to break it into small sentences. For example, A new spatial keyword group query method is proposed in this paper to address the existing issue of user privacy leakage and exclusion of preferences in road networks. The proposed query method is based on the IGgram-tree index and minimum hash set.
1. Introduction
Line 32-34: For a better understanding of the readers, it is suggested to describe the key difference between spatial keyword nearest neighbor query, TOP-k spatial keyword query, the inverse nearest neighbor query and spatial keyword group query.
· Why this paper is focused on “spatial keyword group queries” only?
Line 35-37: The statement, “In recent years, spatial keyword group queries, …… being widely used in daily life and are attracting more and more scholars' attention and research”.
· Please give an example other than that mentioned in Lines 100-102 from daily life where spatial keyword group queries are used
· Why these kinds of queries are getting more attention from scholars/researchers? Do you mean, this is due to large amounts of geo-tagged data being generated by GPS enabled devices or what?
Line 64: It is good to highlight main contributions of this paper. However, there is duplication among the three stated contributions of the paper. Please rectify this accordingly.
Line 88: The statement, “Compared with the previous methods,…..” Please name these methods for clarity
Line 89: Section 2 of this paper is about what? Please briefly describe it, too.
Line 94: Please delete the statement, “The method proposed in this paper is 17% more efficient than the previous method. It is already mentioned earlier.
2. Related Work
It is indeed good to know about brief details of solutions developed by some researchers. However, it is suggested to comment on the limitations of previous solutions. It would certainly add value to your paper.
3. Definitions and Symbol Descriptions
Please describe in simple words, what is differential privacy. As all readers of this paper may not be mathematicians
Line 193: Please define “spatial distance”
Line 195: What is “α”?
Line 200: Statement, “ In this paper, for the convenience of research, α=0.5 is taken” Please give a rational argument for taking α=0.5
4. Query Algorithm Based on the IGgram-tree Index and Minimum Hash Set
· For easy understanding of the readers, it is better to include a flowchart for all pseudocodes e.g. Algorithm 1 to 5.
· Please remove all comments from pseudocodes. E,g. /*Queue PQ queues out an element assigned to G*/
Comments on the Quality of English LanguageExtensive editing required
Round 2
Reviewer 1 Report
Comments and Suggestions for Authors
The authors succeeded in answering my questions.
Reviewer 3 Report
Comments and Suggestions for Authors
Thank you for submitting revised manuscript of better quality.
Comments on the Quality of English LanguageMinor editing is required and can be done during proof reading